# Milk-Derived Carbon Quantum Dots: Study of Biological and Chemical Properties Provides Evidence of Toxicity

**DOI:** 10.3390/molecules27248728

**Published:** 2022-12-09

**Authors:** Hasan Shabbir, Konrad Wojtaszek, Bogdan Rutkowski, Edit Csapó, Marek Bednarski, Anita Adamiec, Monika Głuch-Lutwin, Barbara Mordyl, Julia Druciarek, Magdalena Kotańska, Piotr Ozga, Marek Wojnicki

**Affiliations:** 1Faculty of Non–Ferrous Metals, AGH University of Science and Technology, Mickiewicza Ave. 30, 30-059 Krakow, Poland; 2Faculty of Metals Engineering and Industrial Computer Science, AGH University of Science and Technology, al. A. Mickiewicza 30, 30-059 Krakow, Poland; 3MTA-SZTE “Lendület” Momentum Noble Metal Nanostructures Research Group, University of Szeged, Rerrich B. Sqr. 1, H-6720 Szeged, Hungary; 4Department of Pharmacological Screening, Medical College, Jagiellonian University, Medyczna 9, 30-688 Cracow, Poland; 5Department of Pharmacobiology, Faculty of Pharmacy, Jagiellonian University Medical College, 30-688 Kraków, Poland; 6Technical Secondary School of Chemical and Environmental Protection No. 3, Krupnicza 44, 31-123 Kraków, Poland; 7Institute of Metallurgy and Materials Science of the Polish Academy of Sciences, 25 Reymonta Street, 30-059 Kraków, Poland

**Keywords:** carbon dots, fat-free milk, hydrothermal synthesis, biological applications, toxicity of carbon dot, protein binding, cell survival, antioxidant activity, microwave assisted synthesis, synthesis time-dependent toxicity

## Abstract

Carbon dots (CDs) are carbon-based zero-dimensional nanomaterials that can be prepared from a number of organic precursors. In this research, they are prepared using fat-free UHT cow milk through the hydrothermal method. FTIR analysis shows C=O and C-H bond presence, as well as nitrogen-based bond like C-N, C=N and –NH_2_ presence in CDs, while the absorption spectra show the absorption band at 280 ± 3 nm. Next, the Biuret test was performed, with the results showing no presence of unreacted proteins in CDs. It can be said that all proteins are converted in CDs. Photo luminance spectra shows the emission of CDs is 420 nm and a toxicity study of CDs was performed. The Presto Blue method was used to test the toxicity of CDs for murine hippocampal cells. CDs at a concentration of 4 mg/mL were hazardous independent of synthesis time, while the toxicity was higher for lower synthesis times of 1 and 2 h. When the concentration is reduced in 1 and 2 h synthesized CDs, the cytotoxic effect also decreases significantly, ensuring a survival rate of 60–80%. However, when the synthesis time of CDs is increased, the cytotoxic effect decreases to a lesser extent. The CDs with the highest synthesis time of 8 h do not show a cytotoxic effect above 60%. The cytotoxicity study shows that CDs may have a concentration and time–dependent cytotoxic effect, reducing the number of viable cells by 40%.

## 1. Introduction

Nanomaterials are types of materials that have at least one dimension less than 100 nm. They have unique structures leading to exciting properties which are useful in a number of fields, such as in biology as drug transport or photo luminescent probe for living organism, optoelectronic application like laser diodes, light-emitting diodes (LEDs), solar cells, photodetectors and transistors, electrical applications like quantum computing, data storage and the use of semiconductor nanomaterials to replace conventional semiconductors, and it may also include many more applications in almost every sector of life [1,2]. Nanomaterials can be defined by a number of classifications, and the most important is its dimensions. If all dimensions of nanomaterials are less than 10 nm, it is called a zero dimension. The examples are quantum dot and nanoparticles, and if one dimension is larger than the nanoscale, then they are called 1D nanomaterials. Examples include nanorods, nanowires and nanotubes, and materials with two dimensions that are greater than nanoscale are referred to as 2D nanomaterials. Examples include nanofilms, nanolayers, nano-coatings and quantum wells [3].

Carbon quantum dots (CDs) are carbon-based quantum dots discovered by Scriven et al. in 2004 [4] while working on the purification of single-walled carbon nanotubes. CDs attracted a lot of attention from researchers due to exciting properties such as high photoluminescence, high quantum yield, low toxicity, good water solubility and very good bio-compatibility make them useful for the number of applications. The applications of CDs include antibacterial, florescent probe for labeling, photo catalyst, drug delivery systems, light–emitting diodes, photo–dynamic therapy (PDT) and bio–imaging [5,6,7], etc. Due to a plentiful surface area, CDs can bind with different inorganic and organic molecules to modify the properties by chemical reaction and make them suitable for different applications as compared to traditional semiconductor nanoparticles or quantum dots [8,9]. Nowadays, there are two main areas in research carried out by researchers on CDs. The first is to synthesize the CDs from different low–cost precursors, and the other is to broaden the scope of CDs applications [[10]，[11]]. The phenomena of fluorescence in CDs are not fully known, although the radioactive recombination and excitation of surface defects are considered the cause of photoluminescence. There are several techniques to study photoluminescence behavior in CDs. Wen et al. suggested that there were intrinsic and extrinsic bands [12], while Lei Wang et al. suggest that independent bright molecule-like states are the reason behind photoluminescence [4].

There are a number of precursors that can be used to obtain CDs. Some of them are based on plants and seeds, such as bamboo [13], peanut [14], watermelon [15], sago waste [16], walnut [17], osmanthus fragrans [18], strawberry [19], and banana [10]. Organic compounds like urea, ascorbic acid, and glucose are also reported as a precursor for CDs [10].

Many researchers have already conducted a study on the impact of CDs generated through various methods and from diverse precursors. Janus et al. assessed carbon quantum dots for their biocompatibility with human skin fibroblasts to verify their safety in biomedical applications and their potential in diagnostics and bio-imaging [20]. It was observed that the toxic effect on primary cells is correlated with the concentration of CDs; however, cell survival in no case fell below 80%. There were also no abnormalities in the morphology of fibroblasts exposed to quantum carbon dots for 48 h.

CDs are generally considered nontoxic and more biocompatible than their metallic counterpart [21]. One study shows the high dose of CDs is about 2 g/kg derived from cola when injected into mice does not cause death [22]. The CDs did not cause toxicity in cells at concentrations as high as 20 mg/m when reacted directly to the cell of small guppy fish [[23]，[24]]. CDs generated from organic carbon sources can cure cancer cells as effectively as conventional medications while causing no harm to healthy cells. CDs mostly possess properties similar to their precursor [25].

Nanomaterials can be synthesized using a variety of techniques, including laser ablation, ultrasonic treatment, chemical oxidation, hydrothermal heating, and microwave heating [[26]，[27]]. Each of these techniques has its pros and cons. Microwave heating has received a lot of attention because it allows for fully controlled temperature and pressure, which establish the properties of nanomaterials. The majority of approaches involve increasing the temperature to complete the reaction [28]. There will be some hydroxides left in the nanomaterial if the reaction temperature is low, and this can alter the properties of nanomaterials. The microwave method is becoming more and more popular because of its inexpensive cost for equipment modification, rapid and uniform heating, and a high degree of purity. We used the Ertec magnum reactor, which was specially made to manufacture nanomaterials, to produce carbon dots for the first time with high precision. This reactor can produce and sustain a pressure of 6 Mpa and temperatures of 260 °C [29].

To the best of our knowledge, CDs toxicity and its influencing factors have not been thoroughly studied. To explore the toxicity of CDs, we used mouse hippocampal cells and synthesized CDs from fat-free milk in this work to prevent the complex chemical formation at CDs shell from fat. Surprisingly, these CDs exhibit significant toxicity, especially for 1 and 2 h synthesis time.

## 2. Results

This section may be divided by subheadings. It should provide a concise and precise description of the experimental results, their interpretation, as well as the experimental conclusions that can be drawn.

### 2.1. Ultraviolet–Visible and Photo–Luminescence Spectroscopy

UV and PL spectra were measured to find the absorption and emission peak of CDs. The results are shown in Figure 1. The CDs were synthesized for 1, 2, 3, 5 and 8 h (h) and labeled as K1 h, K2 h, K3 h, K5 h and K8 h, respectively, for all the figures in this research article.

The absorption of the materials should be known to scan the material for excitation. The excitation peak is always higher than absorption due to loss of energy in the intersystem crossing (ISC), which does not contribute to the emission and excitation process but is involved in other energy loss processes such as heating. The Figure 1A shows that CDs have a peak near to 280 ± 3 nm that is attributed as π-π* transition of conjugated C=C and a non–covalent weak n-π* transition of C=O bonds, respectively. The CDs was further excited at wavelength higher than 80 nm that is from 300 nm. The Figure 1B shows the photoluminescence spectra for CDs. There is no established relation between the synthesis time and photoluminescent intensity; however, for the 3 h synthesis time, the intensity is highest and shows the formation of CDs are complete after three h and after this structure begins to break. The position of PL peak does not change with synthesis time. This suggests that the size and chemical properties of CDs are independent to the synthesis time.

### 2.2. Fourier–Transform Infrared Spectroscopy (FTIR) and X-ray Powder Diffraction (XRD)

FTIR analyses were performed for solid samples. This analysis allows for the investigation of the presence of functional groups at the surface of CDs. Obtained results are shown in Figure 2.

After utilizing FT–IR spectroscopy to investigate the functional group on precursor as well as CDs, the results clearly demonstrated changes in the peak after the reaction, as shown in Figure 2. For the milk, all peaks are large in comparison to CDs, which showed a decrease in the intensity of organic bonds after the hydrothermal reaction. The peak at 890 cm^−1^, represents a triazine ring mode, which matches the condensed CN heterocycles mentioned in the literature [30]. A sharp peak was present at 1060 cm^−1^, which shows C-O group, and this peak is not present for the CDs. At the 1666 cm^−1^ wavelength, the C=O bond peak was present. After the hydrothermal reaction only the intensity changed; its wavelength remained unchanged while the small peak band near 1247 cm^−1^ can be attributed to C-N stretching. A peak near 1415 cm^−1^ confirms that C=N stretching-H stretch can be observed at 3300 cm^−1^. The peak at 2920 cm^−1^, which represents C-H, also changed its wavelength position and also shows changes in bonding. The O-H (acid) peak also changes its wavelength position [31]. From this it can be concluded that CDs have many carboxyl and hydroxyl group, caused by oxidation, and nitrogen containing functional groups such as the amine group due to protein in the milk.

Figure 3 depicts the CDs diffraction pattern, with the resolvable peak being visible at 22.5°, which is similar to the graphite diffraction peak at 26.125° [32]. While the peak at 30° shows the presence of NaCl crystals in the milk precursor, which is similar to other findings in the literature [33]. The CDs peak indicates that the interlayer spacing is larger than graphite and also confirms the poor crystalline nature of CDs.

### 2.3. Detection of Protein by Beirut Test

As zero fat milk was used as a precursor for CDs, which contains carbohydrates, sugar and protein. All of these ingredients can be used as precursor for CDs. To calculate the production yield, of CDs following procedure was applied. First, a drying process of 1 mL of CDs and milk suspensions was done at 80 °C overnight. Next, the dry mass was measured. The dry mass concentration for milk is c.a. 90.9 g/L. In the case of CDs, as expected, the dry mass was found lower in comparison to precursor. Compering initial concentration of dry mass in milk to concentration of dry mass of obtained CDs, product yield was calculated. It was found that it decreases after the reaction and for the 8 h synthesis time the dry mass of CDs is lowest, which is 27.7 g/L, but there is no direct relationship between the reduction of mass and synthesis time established. The complete detail of mass after drying can be seen in Table 1.

Based on the results of this preliminary study, a Beirut test was conducted to investigate whether there is any protein left over after the reaction that does not convert into CDs. First, the calibration curve was made using serum bovine albumin as a reference protein (see Figure 4A,B). Next, a sample of CDs and milk were treated in the same way, and the UV–Vis spectra were recorded. The obtained results are shown in Figure 4C. The CDs does not change its color and remains pale yellow, while the milk changes its color to purple, which shows there are no proteins present in CDs.

In the case of CDs, insignificant lift up of the UV–Vis spectrum at 547 nm is observed. This is related to the light scattering caused by the CDs. In this region. no peak was observed. This is a confirmation that the milk proteins were fully converted to the CDs.

### 2.4. High-Resolution Transmission Electron Microscopy and Raman Spectroscopy

A high-resolution scanning transmission electron microscopy (HR–STEM) study was carried out to determine the size and size distribution of CDs. HR–STEM analysis was performed for the samples obtained after 2 and 8 h of synthesis time. The obtained results are shown in Figure 5A,B.

Based on the TEM images, the size distribution was calculated. The obtained results are shown in Figure 5C,D. As it can be seen, in the case of CDs synthesized for 2 and 8 h, the average size and size distribution are comparable. The main fraction is c.a. 2 nm in diameter.

In the Figure 6A,B Raman spectra were carried out on gold layers (approx. 150 nm) deposited on atomically planar mica structures. After evaporation of the dispersing solvent, the carbon fragments merged into larger structures. Raman studies indicate the presence of carbon amphora structures below 1 nm. The presence of this type of structures is indicated by the G band of sp^2^ structures [34]. The width of the G band, for which FWHM is about 94 cm^−1^, and the absence of the D band showed a significant fragmentation of sp^2^ structures, which thus indicates the presence of amorphous sp^2^ carbon structures of considerable size (from tens of nanometers to micrometers) that show much a smaller broadening band G band of the order of approx. 16–20 cm^−1^).

### 2.5. Dynamic Light Scattering (DLS), Zeta Potential Analysis and Atomic Force Microscopy

CDs surface charges were determined by zeta potential measurements of −14.9 mV while milk has a zetapotential value of −7.27 mV. The conductivity value of the solution containing CDs was increased from 5.31 mS/cm for milk to 13.9 mS/cm for CDs. This is due to the destruction of functional group of milk during hydrothermal reaction. Hydrodynamic diameter was measured and was equal to 3.0 nm, which is slightly larger than determined using HR–STEM. This is due to the size of the liquid layers around the CDs in the case of DLS measurements. Figure 7A,B illustrates the hydrodynamic diameter of milk and CDs, which reveals that the milk contains big particles with a size greater than 100 nm, while for CDs the particle size is in the range of 2–4 nm.

This measurement also confirms that the proteins were destroyed during the CDs synthesis. Since large particles were not observed in CDs suspension.

Figure 8 shows an atomic force microscopy image of the agglomeration of CDs after evaporation of solvent suspended on HOPG substrate.

### 2.6. Bovine Albumin

The bovine albumin protein is obtained from cows and has a very large molecular structure. The binding of bovine albumin with CDs was tested. Two solutions were prepared. The first solution of 10 mL of 0.00025 M bovine albumin was mixed with 3 mL of CDs. Then, a dialysis test was conducted on the mixture of these solutions for 24 h. The CDs passed through the dialysis membrane. Bovine albumin solution did not pass since the cut-off of the membrane was preselected to avoid this process. This simple experiment confirms that there is no interaction between bovine albumin and CDs or the interaction is weak. Next, the bovine albumin solution and CDs suspension was analyzed with the help of UV spectroscopy and the results are shown in Figure 9. For this purpose, we used tandem cuvette. The spectra before and after mixing the bovine albumin solution with the CDs are shown in Figure 9.

There was a degree of change in the absorption spectra, which confirms that there is a weak interaction between bovine albumin and obtained CDs. Based on these simple experiments, we can expect that obtained CDs will also show low or no toxic effects. However, due to the compactivity of live cells, further studies are required.

### 2.7. Antioxidant Assays and Toxicity Study—In Vitro

#### 2.7.1. DPPH Assay

The DPPH test is used to assess antioxidant activity using the free radical scavenging mechanism. As a reference compound with antioxidant properties, ascorbic acid was used in the concentration range of 0.1 to 1 mM, which, at the highest concentration, showed the maximum antioxidant effect (100%) expressed as a decrease in absorbance at a wavelength of 517 nm.

All types of CDs showed comparable concentration-dependent antioxidant activity. The CDs at the highest concentration used, that is 4 mg/mL, showed antioxidant activity in the DPPH test at a level of 63% to 77% of the maximum ascorbic acid activity, and their activity decreased linearly with decreasing concentrations, while those with shorter synthesis times of 1 and 2 h turned out to be the most active (Figure 10).

#### 2.7.2. FRAP Assay

FRAP assay measures the reducing potential of an antioxidant reacting with a ferric tripyridyltriazine (Fe^3+^–TPTZ) complex and producing a colored ferrous tripyridyltriazine (Fe^2+^–TPTZ).

In the in vitro determination of total antioxidant activity (FRAP), ascorbic acid was used as a reference compound. It has been shown to be a proportional increase dependent on concentration (0.1–1 mM) proportional increase in absorbance resulting from an increase in total antioxidant activity determined by the concentration of reduced iron (Figure 11A). The highest activity of ascorbic acid was demonstrated for a concentration of 1 mM.

Ascorbic acid was used as an antioxidant reference compound in this test and produced a concentration-dependent linear reduction of 1 mmol Fe^2+^ per 1 mmol of ascorbic acid at the highest concentration tested.

In the FRAP test, all types of CDs—K1 h, K2 h, K3 h, K5 h and K8 h—caused a concentration-dependent reduction of iron (III) ions. Carbon dots with shorter synthesis times, i.e., 1 and 2 h, showed higher antioxidant activity in this test. At the highest concentration tested, i.e., 4 mg/mL, the most active dots reduced approx. 1.8 mM Fe^3+^, while the least active dots, i.e., those synthesized for 5 and 8 h, reduced iron ions (III) at a concentration of approx. 1.2 mm (Figure 11B).

#### 2.7.3. Lipid Peroxidation

Free malondialdehyde (MDA) is a marker of lipid peroxidation and one of the markers of oxidative stress. It is formed as a result of the reaction of reactive oxygen species with unsaturated fatty acids and their subsequent decomposition into aldehydes. The lipid peroxidation test is based on the quantification of the malonyldialdehyde formed, which is capable of reacting with thiobarbituric acid to form colored adducts. Carvedilol in the concentration range of 10^−7^ to 10^−3^ M was used as a reference compound, for which the calculated EC_50_ value was 28.69 µM and the results are shown at Figure 12.

At a concentration of 2 mg/mL all tested CDs showed antioxidant activity at a level of 100% carvedilol activity (10^−3^ M) in a concentration 10 times lower, i.e., 0.2 mg/mL, though it decreased by approximately 50–65%, regardless of their nature.

#### 2.7.4. Cytotoxicity

Among the tested types of CDs, at the highest concentration used, i.e., 4 mg/mL, all showed cytotoxic activity on HT22 cells (mouse hippocampus cells). At this concentration, the dots with the shortest synthesis time; that is, 1 and 2 h, showed the strongest cytotoxic effect, and the fluorescence intensity, reflecting cell survival, was less than 20%, as shown in Figure 13. Reducing their concentration by half drastically weakened the cytotoxic effect, ensuring survival at a level of 60–80%. Other types of CD, with longer synthesis times (3, 5 and 8 h), given at a concentration of 4 mg/mL, also decreased cell viability, but in their case, cell viability was approximately 40% (moderate cytotoxic effect), and concentration reduction also decreased the cytotoxic effect (higher survival). However, for the longest synthesized dots, i.e., 8 h, the viability of the cells, even at the lowest concentration, did not exceed 60%.

## 3. Discussion

In this paper, the synthesis of milk base CDs was carried out. The obtained CDs exhibit a narrow size distribution (±1 nm) and the size is independent of the synthesis time (d = 3 nm). However, the surface chemistry of these CDs depends on the synthesis time. This was noted in the FT–IR spectra.

Taking into account the promising data from the literature on the potential antioxidant activity, FRAP and DPPH tests were also carried out for the quantum carbon dots, being the subject of this study, to determine the total antioxidant activity in vitro and to determine the effect on lipid peroxidation in vitro using the MDA method.

FRAP and DPPH test confirms that CDs show antioxidant activity, which depends on their concentration and the time of their synthesis. In the FRAP study, CDs with a concentration of 4 mg/mL, CDs with the shortest synthesis time (K1 h and K2 h) were the most active because they reduced by approximately 1.8 mM Fe^3+^, while the least active dots (K5 h and K8 h) reduced the Fe^3+^ ions at a concentration of approximately 1.2 mM. The activity in the FRAP test may result from both antioxidant properties and the chelating effect of iron III ions. Taking into account the FT–IR results, the difference between K1 h and K8 h is clearly seen. This confirms that surface chemistry (types and concentration of functional groups) has a significant impact on the antioxidant properties.

The antioxidant activity of the tested compounds was confirmed in the DPPH test, in which the CDs with the highest concentration—4 mg/mL—showed the antioxidant activity at the level from 63% to 77% of the maximum activity of ascorbic acid used as the reference compound. The obtained results are consistent with the data obtained from the literature for other types of quantum carbon dots. Janus et al., in their study of carbon dots obtained on the basis of a L–lysine polymer exposed to microwave radiation [20], showed that the antioxidant activity of quantum carbon dots strictly depends on their type, and the dots with the shortest synthesis process have a stronger antioxidant effect because prolonging the synthesis time reduces the number of surface functional groups leading to the formation of a carbon inactive core [20].

When determining the effect on lipid peroxidation in vitro using the MDA method, CDs at a concentration of 2 mg/mL showed antioxidant activity at the level of 100% of carvedilol activity (10^−3^ M), and at a concentration 10 times lower, i.e., 0.2 mg/mL, it decreased by about 50%, regardless of their type. The above test results indicate the antioxidant potential of the tested materials, which depends on the concentration and, in some cases, the time of their synthesis.

Studies of carbon dots differing in the method of synthesis also showed that, practically, regardless of their size in the concentration range of 100–400 µg/mL, they did not exhibit cytotoxic activity, and cell survival after 1, 4 or 7 days was around 90 [36].

However, it still seems important to study the intracellular transport and distribution of quantum carbon dots, especially transport to the nuclear area of the cell. These parameters will certainly be influenced by the completion of the surface of carbon quantum dots. Penetration of nanoparticles into the nucleus is limited by the pore diameter; however, particles below 50 kDa pass through the nuclear membrane in both directions in the simple diffusion mechanism using water channels with a diameter of about 9 nm [37]. As shown by the research, quantum carbon dots up to a concentration of 400 mg/mL did not significantly affect the viability of cells or the content of DNA in them (assessing genotoxic effect). As a result of further study, it was found that quantum carbon dots accumulate in the cell nucleus, and when given in high concentrations, they might lead to morphological changes in the cell, especially near to the nucleus, and ultimately causes death of the cell. However, this effect depends on the type of cells tested, and the strongest effect cytotoxic and genotoxic have been observed on the L929 line.

Shi et al., by examining various types of carbon nanostructures, showed that quantum carbon dots did not show cytotoxic effect with the MTT test on A549 (type II pneumocytes) and HACAT (keratinocytes) cell lines at concentrations up to 1.333 mg/mL. Only for the concentration of 4 mg/mL was a decrease in the survival of A549 cells observed below 40% and HACAT up to 80% [38]. In the same study, other carbon nanostructures, such as graphene spheres and nanotubes, showed greater cytotoxicity observed at much lower concentrations.

Tian et al. also investigated the safety of carbon dots both in in vitro cell cultures and in vivo in mice. Studies with NP69, CNE2, HepG2, and RAW264.7 cell lines did not show significant differences in survival compared to the negative control. Apoptosis was assessed on the NP69 and RAW264.7 cells by flow cytometry. For quantum carbon dots at the concentration of 10 mM, no significant differences were found compared to the negative control, which proves the lack of apoptotic effect. Studies of the cellular distribution of CQDs showed their presence in the cytoplasm and cell nucleus but without affecting the integrity of cell membranes. In vivo immunotoxicity studies in mice did not show any effect on ROS levels in immune cells and gene expression (PI3K, P38, HSF1, JNK, ERK, P53, and NF–κB1) associated with liver immunotoxicity, such as after carbon dots injection [39].

In summary, the results obtained show that carbon quantum dots synthesized from 0% milk fat have significant antioxidant activity, which depends on their concentration and synthesis time. The materials tested show an acceptable safety profile in terms of cytotoxicity at low concentrations. The above results suggest the possibility of conducting in vivo tests at a later stage of the research.

## 4. Materials and Methods

### 4.1. CDs Synthesis and Characterization

Fat-free cow milk was used as precursor for CDs to investigate the role of protein on CDs. The hydrothermal method was used for this purpose. In the standard experiment 40 mL of milk was heated in a closed Teflon made reactor and the temperature of the reaction was in the range of 190–200 °C. The Teflon-made reactor is a part of the Ertec Magnum II reactor (Ertec, Poland). The reactor has a filling capacity of 108 cm^3^ and its maximum operational pressure is 50 bars, while the maximum operational temperature that can be achieved is 300 °C.

After synthesis, a dark yellow (close to brown) color CDs was obtained and which changes its color to pale yellow after dilution. The reaction was carried out for different durations of 1, 2, 3, 5 and 8 h to study the effect of the synthesis time of CDs on their properties. A pale–yellow liquid obtained after the reaction was filtered. For this purpose, a syringe filter (200 nm) was used to remove the large particles from the liquid.

The hydrodynamic diameter and zeta potential of CDs was measured with a Zetasizer Nano ZS (Malvern Instrument, Malvern, UK). This instrument is able to measure the particle size and zeta potential in dispersed solution; therefore, a dilution of CDs in water was made for this purpose. The measurement was done in a standard transparent polycarbonate cell with gold electrodes.

Horiba FluoroMax 4 spectrometer (HORIBA Europe GmbH—Dresden Office) was used to determine the photoemission properties of CDs. The standard for walls transparent cuvette with 1 cm optical length was used. This spectrometer has calibrated xenon light, which provides UV light that is utilized during measurement.

Shimadzu model U–2501 PC spectrophotometer (Shimadzu corporation, Kyoto, Japan) was used to record UV–Vis spectra. The spectra can measure the UV–Vis in range of 190–900 nm wavelength. The baseline was corrected with water for measuring only absorption spectra and when measuring the quantum yield, it was corrected by sulphuric acid, which was used as a solvent for quinine sulphate.

Nicolet 380 spectrometer (Thermo Fisher Scientific, Bremen, Germany) was used to measure the FT–IR spectra to identify the functional groups. Dry spectroscopic grade potassium bromide (KBr) was used as a reference. The KBr was mixed with each sample in the proportion of (0.2 g KBr ± 0.0002 g of sample) and pellets were made with the help of hydraulic press. These pellets subsequent analyzed using FT–IR.

The dialysis method was used to analyze the bonding of CDs with bovine albumin protein and it carried out for more than 24 h. The dialysis tube had the following specifications: wall thickness 0.03 mm, diameter 28.6 mm, and length 30 m.

Titan^3^ G2 60–300 (Thermo Fisher Scientific, Eindhoven, The Netherlands) in scanning transmission electron microscopy (STEM) mode was used for microstructure characterization. The high–angle annular dark field (HAADF) technique was utilized in order to depict the structure of the CDs. It was the most suitable method in order to detect carbon CDs because the contrast depends on both the atomic number and the thickness of the investigated particle. It has key importance in the present study as the couple nm–sized carbon CDs were dispersed in ethanol and deposited on a standard TEM grid (~35 nm thick amorphous carbon film on copper mesh). In such conditions, obtaining any contrast is very difficult. Before investigation, samples were cleaned with plasma (Plasma Cleaner, Model 1020, Fischione instruments) for 2 s, however, the contamination was not fully avoided. In order to minimize irradiation damage of the sample, most of the investigated area was scanned only once with very low beam current (~40 pA); therefore, the presented results are affected by the noise.

The image from HR–STEM was analyzed using ImageJ version 1.52v. Images from the microscope were analyzed using a several-step protocol related to image preparation. First, the scale was registered. Next, the image was cut to remove the scale bar and text box. Next, the image was filtered using FFT band–pass filter. The settings of the filter were as follows: filter large structure down to 40 pixels, filter small structures up to 3 pixels, suppress stripes none, tolerance of direction 5%. Next, the particle size was determined as follows: using threshold set up, carbon quantum dots were detected, and image was converted to black–white. Using a particle analysis tool, the surface areas of carbon quantum dots were determined. The following parameters were selected: size nm2 from 1 to infinity. This allows removing single pixels and artifacts from image. Next, circularity was selected in the range 0.5–1. This allows excluding any particles from the analysis that are stuck together. At the end, the surface of single carbon quantum dots was determined, the diameter was calculated, and the histograms were plotted. Raw dataset fails as well as after post-processing are available at the date depository [40].

### 4.2. Antioxidant Assays—In Vitro

Antioxidant properties of investigated CDs were tested in vitro in two different assays: the 2,2–diphenyl–1–picrylhydrazyl (DPPH) assay [41,42,43] and the FeCl_3_ activity reduction assay (FRAP, ferric reducing antioxidant power) [42,43]. The tested CDs were dissolved in ethanol or in water (respectively) at concentration of 4 mg/mL or 10−2 M and then diluted with ethanol.

#### 4.2.1. DPPH Assay

Measurements were performed according to Brand–Williams et al., with some modifications [44]. The assay is based on the measurement of the scavenging capacity of the antioxidants towards it. The odd electron of a nitrogen atom in DPPH is reduced by receiving a hydrogen atom from the antioxidants to the corresponding hydrazine [45]. The antioxidant activity of tested CDs was determined by spectrometric determination of the reduced form of DPPH as percent of maximal antioxidant activity of ascorbic acid (1 mM).

A solution of a stable free radical, DPPH (1,1–diphenyl–2–picrylhydrazyl; Sigma–Aldrich, Germany) in concentration 200 µM/L in ethanol was stored in the dark for 2 h. Solutions of selected CDs in ethanol at the concentration of 4 mg/mL were prepared and diluted with ethanol to obtain concentrations: 0.125, 0.25, 0.5, 1, and 2 mg/mL. To 20 µL of the tested CDs or solvent, 180 µL of the DPPH solution were added. The mixtures were vortexed and allowed to stand for 20 min in the dark at room temperature (25 °C). Absorbance was measured at 517 nm against ethanol as a blank. Figure 14 represent % of ascorbic acid antioxidant effect.

#### 4.2.2. FRAP Assay

The assay was performed according to Benzie and Strain with some modifications [46]. The total antioxidant activity of the tested compounds was determined by spectrophotometric determination of the reduced iron concentration. In the experiment, to 10 µL of the test CDs or solvent, 300 µL of the reagent were added with the following composition: 10 parts of a 0.3 M sodium–acetate buffer, pH 3.7, 1 part of 0.01 M TPTZ solution, 1 part of 0.02 M FeCl_3_ × 6 H_2_O solution. Absorbance was measured after 10 min incubation at room temperature at 593 nm against ethanol as a blank. The results for the tested CDs were presented as the amount of reduced iron (III) ions, as the reference compound, ascorbic acid was used. The antioxidant capacity of which was determined in the concentration range from 100 to 1000 µM. Deionized water with FRAP solution was used as a blank.

Next, FeSO_4_ × 7 H_2_O salt was used for standard curve construction (Figure 15). For this purpose, the salt was dissolved in water and further diluted to obtain concentrations of 0.025 to 2 mM. From the CDs, 4 mg/mL ethanol solutions were prepared and diluted with water to obtain concentrations: 0.125, 0.25, 0.5, 1, and 2 mg/mL.

#### 4.2.3. Lipid Peroxidation

The antioxidant activity was assessed on the basis of the measurement of lipid peroxidation in rat brain homogenate. The rat brain homogenate was made in 0.9% saline containing 10 mg of tissue per mL. The rates of membrane lipid peroxidation were measured by the formation of thio barbituric acid reactive substances (TBARS). Rat brain homogenates (1 mL) were incubated at 37 °C for 5 min with 10 µL of a CDs, test compound or vehicle. Lipid peroxidation was initiated by the addition of 50 µL of 0.5 mM FeCl_2_ and 50 µL of 2 mM ascorbic acid. After 30 min of incubation, the reaction was stopped by adding 0.1 mL of 0.2% BHT. Thio barbituric acid reagent was then added and the mixture was heated for 15 min in a boiling water bath. The TBARS was measured at 532 nm. The amount of TBARS was quantified using a standard curve of MDA. 1,1,3,3TEP was used as the MDA standard.

### 4.3. Cytotoxicity Assay

To assess the cytotoxicity effect of the compounds, a Presto Blue test was performed. An experiment was carried out on a POLAR star Omega, plate reader (BMG Labtech). Cell viability was measured using the Presto Blue reagent (Invitrogen). Presto Blue reagent is a resazurin–based solution that functions as a cell viability indicator is used. HT22 cells (mouse hippocampal cells) were used for the assay, 2000 per well in DMEM Glutamax with 10% FBS and 1% Pen/Strep solution. Metabolically active cells are capable of reducing the Presto Blue reagent, with the colorimetric changes used as an indicator to quantify the viability of cells in culture. This change can be determined by measuring the fluorescence. After 24 h of incubation with the tested compounds, a Presto Blue reagent was added to wells of a microplate in an amount equal to one tenth of the remaining medium volume. After 15 min of incubation at 37 °C, the fluorescence intensity (EX 530; EM 580 nm) was measured in a plate reader. Viability values were calculated as a percentage of live cells with respect to the control sample (DMSO). Staurosporine (known cytotoxic compound) was used at concentration of 0.1 mM to show cytotoxic activity. The negative control was medium without cells.

## Figures and Tables

**Figure 1 molecules-27-08728-f001:**
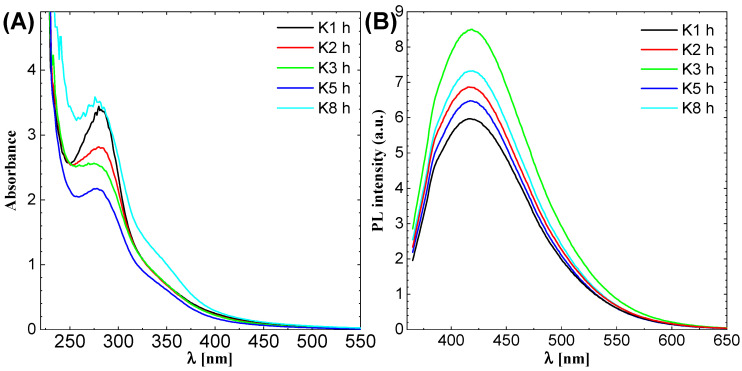
(**A**) Ultra-Violet spectroscopy and (**B**) Photo–Luminescence Spectroscopy of CDs.

**Figure 2 molecules-27-08728-f002:**
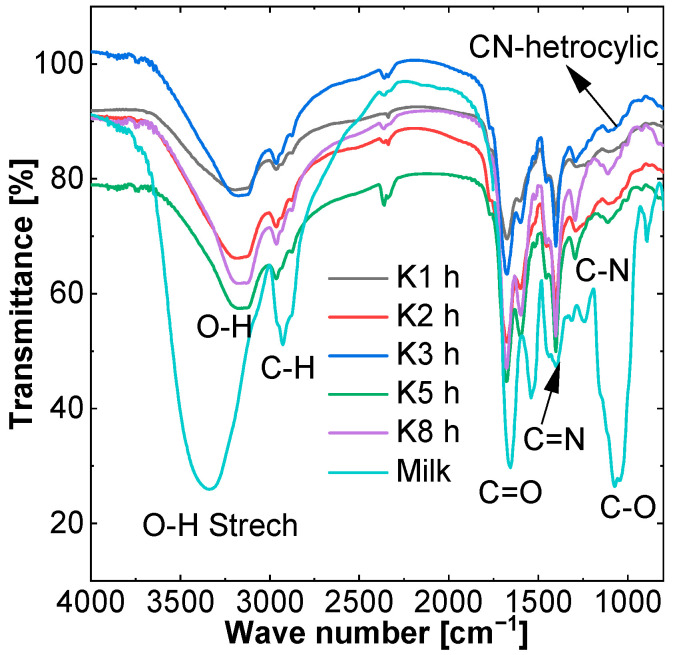
FTIR of milk-based carbon dots and dry milk.

**Figure 3 molecules-27-08728-f003:**
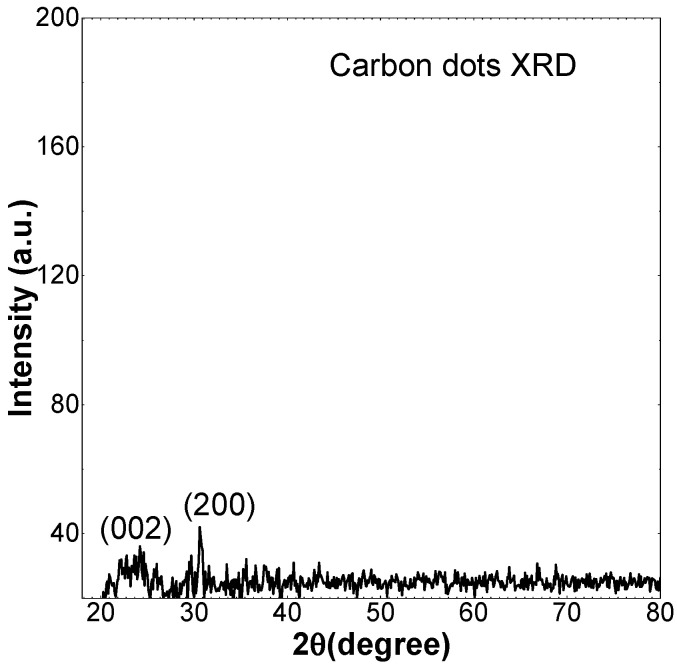
X-ray Powder Diffraction (XRD) of Carbon dots.

**Figure 4 molecules-27-08728-f004:**
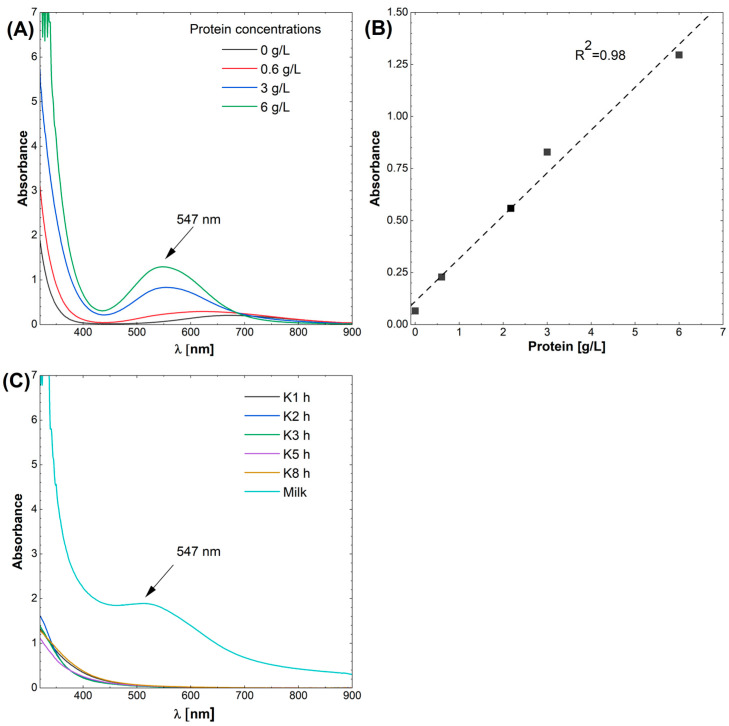
(**A**) UV–Vis spectra of the solution after biuret reaction for different initial concentrations of protein, (**B**) determination of molar absorption coefficient, (**C**) analysis of CDs.

**Figure 5 molecules-27-08728-f005:**
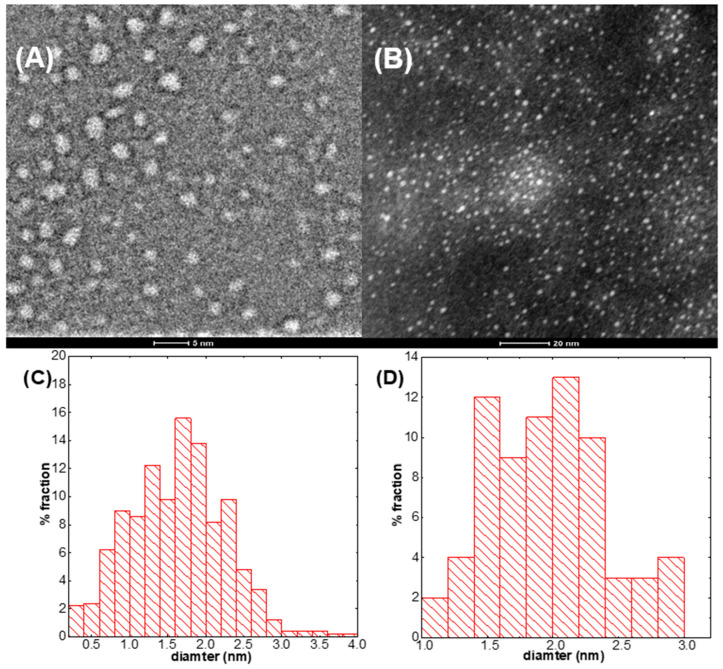
(**A**) TEM image of CDs after 2 h synthesis, (**B**) TEM image of CDs after 8 h synthesis, (**C**) size distribution for CDs after 2 h synthesis (K2 h) and (**D**) size distribution after 8 h synthesis (K8 h).

**Figure 6 molecules-27-08728-f006:**
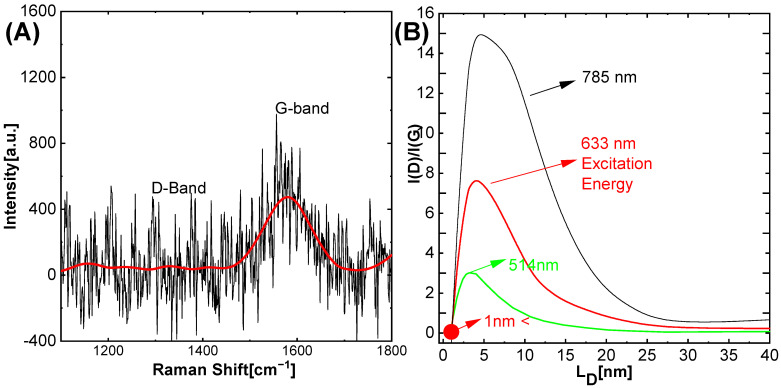
Raman Spectra of Carbon dot (**A**) Represents the D and G band. (**B**) Represents excitation energy based on literature [35].

**Figure 7 molecules-27-08728-f007:**
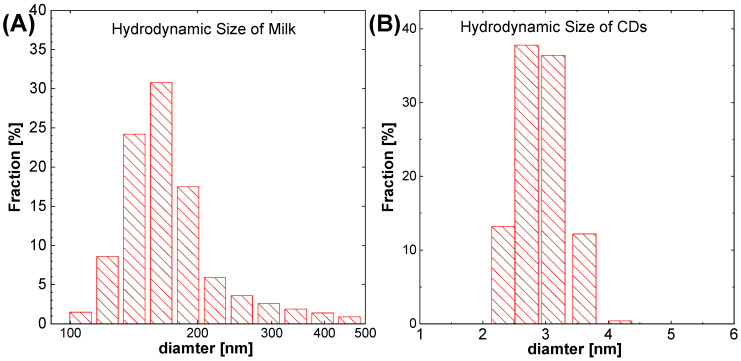
(**A**) Hydrodynamic size of milk proteins, and (**B**) Hydrodynamic size of CDs for 8 h synthesis time with dynamic light scattering method.

**Figure 8 molecules-27-08728-f008:**
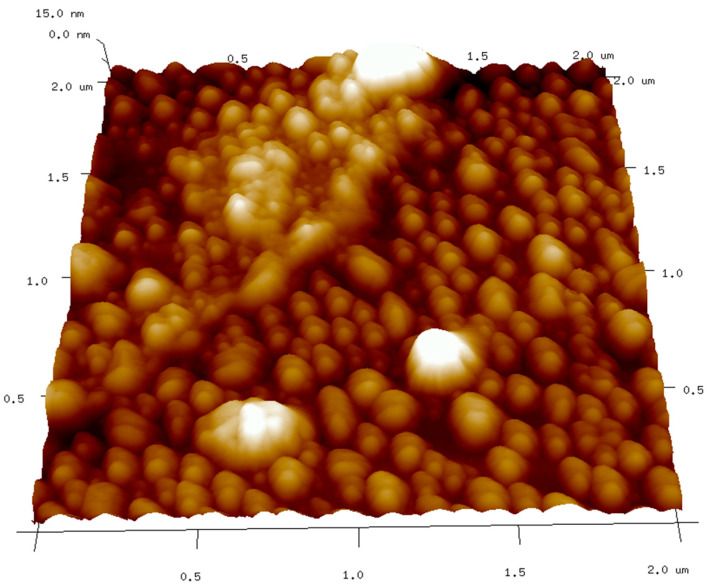
Atomic force microscopy image of carbon nanodots agglomerates.

**Figure 9 molecules-27-08728-f009:**
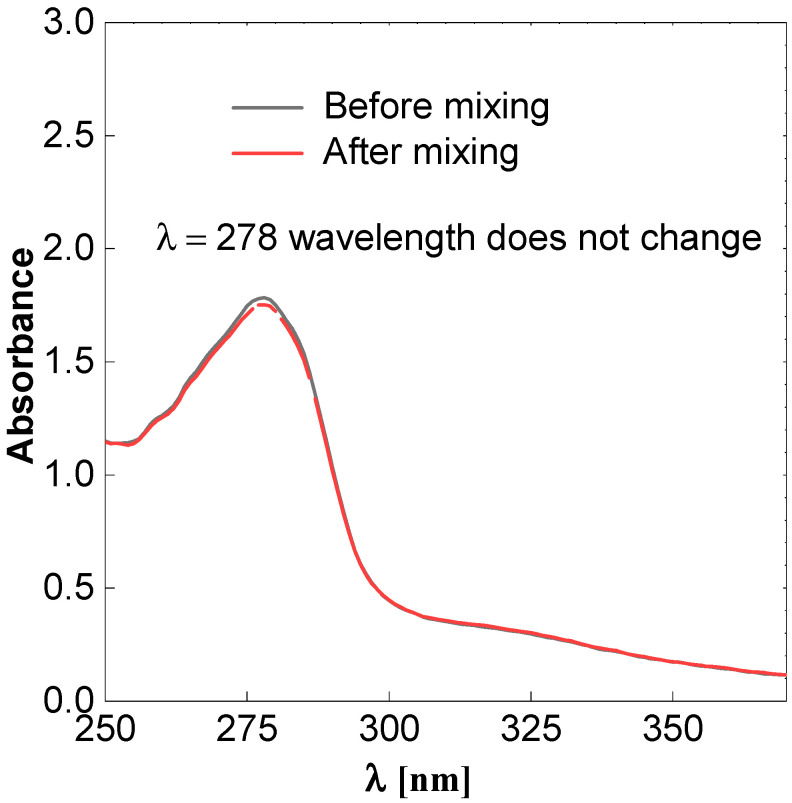
UV-Vis spectroscopy of bovine albumin before and after mixing carbon dots.

**Figure 10 molecules-27-08728-f010:**
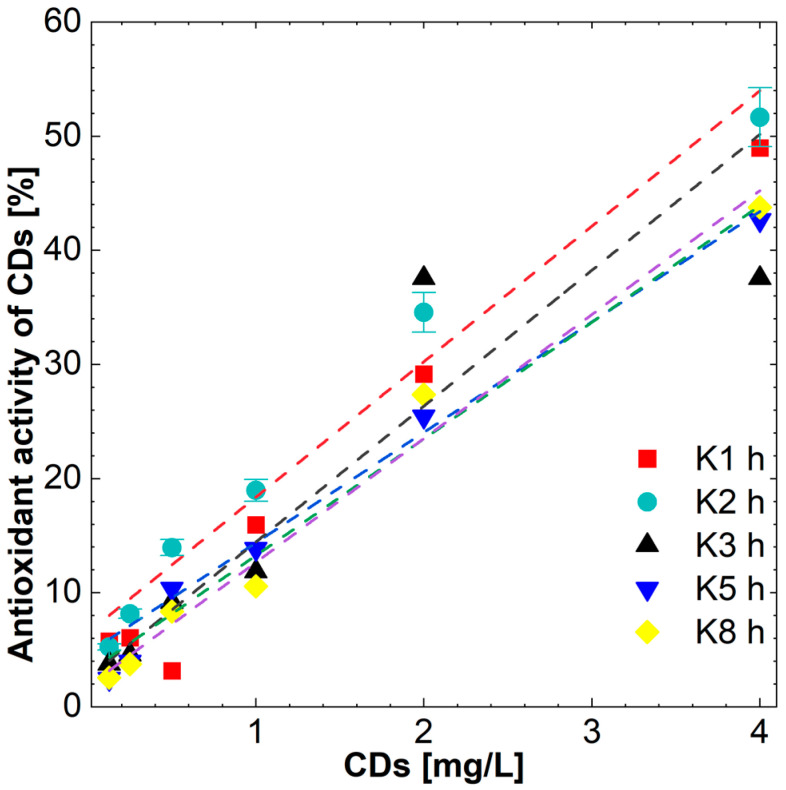
Antioxidant effect of CDs in DPPH assay.

**Figure 11 molecules-27-08728-f011:**
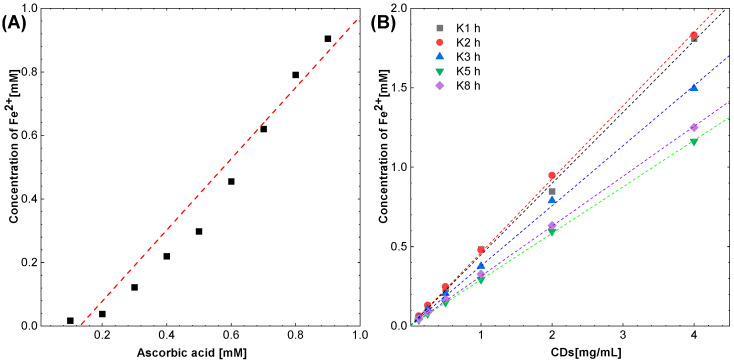
Antioxidant effects of (**A**) ascorbic acid and (**B**) CDs in FRAP assay.

**Figure 12 molecules-27-08728-f012:**
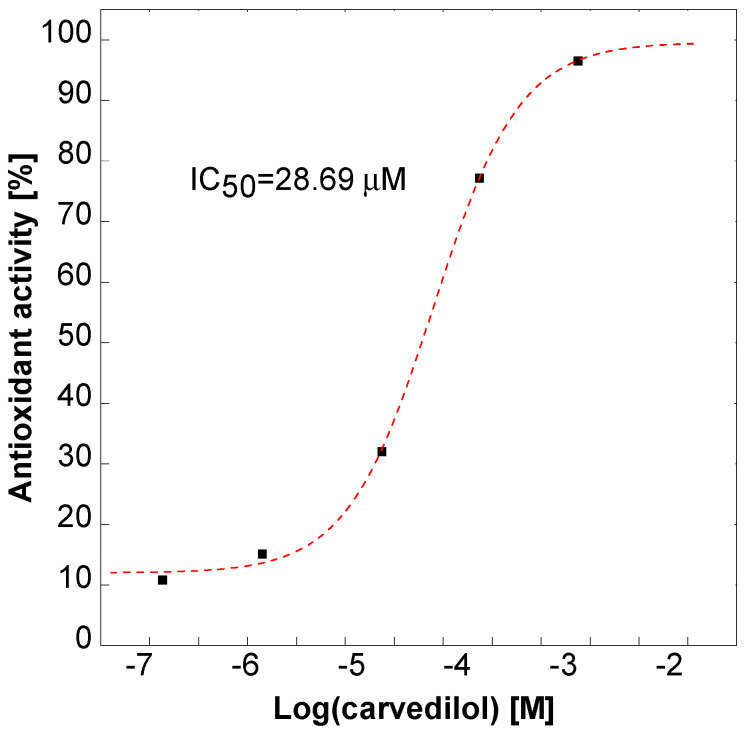
Percentage of antioxidant activity (inhibition of lipid peroxidation) of the reference compound—carvedilol.

**Figure 13 molecules-27-08728-f013:**
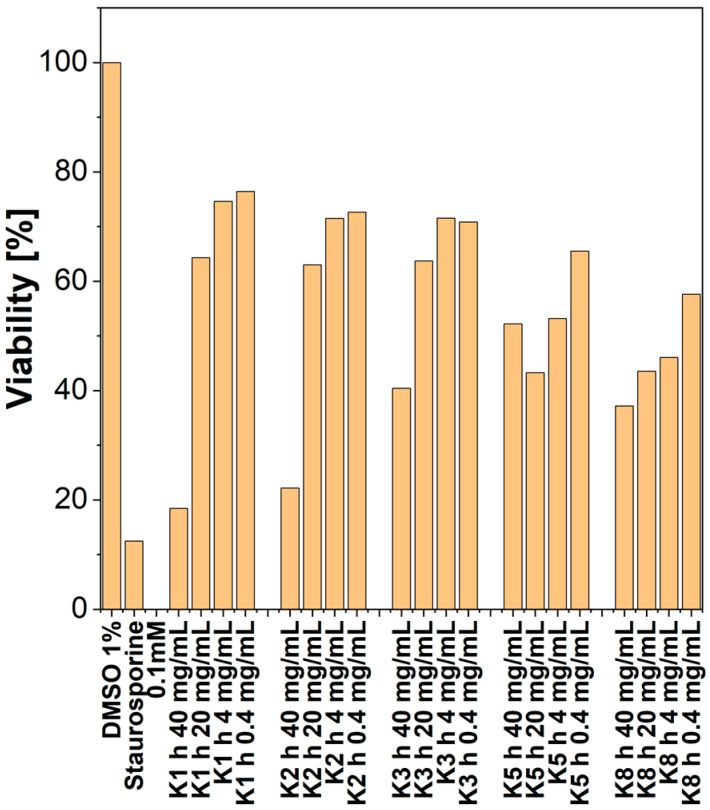
Cytotoxicity and synthesis time of CDs.

**Figure 14 molecules-27-08728-f014:**
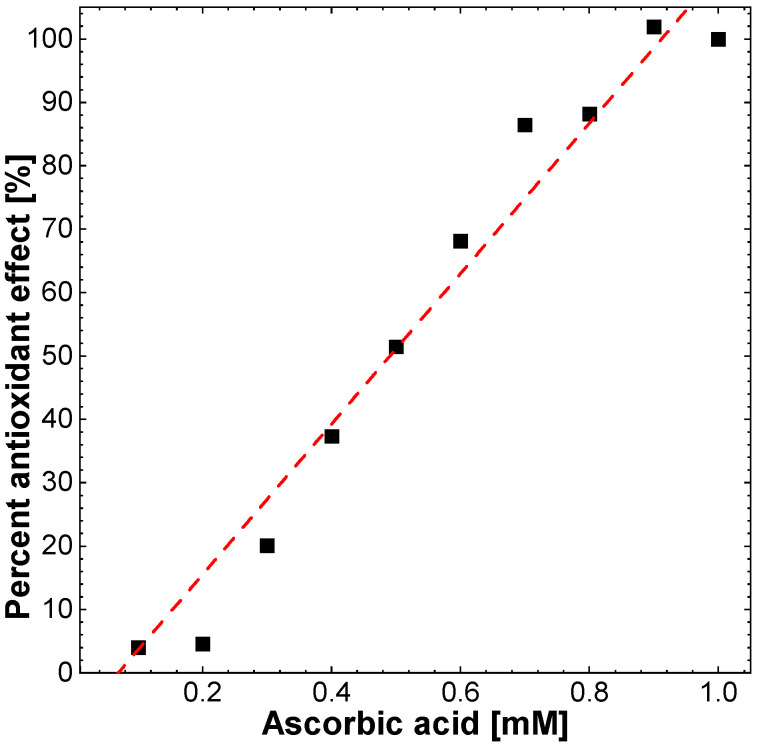
Antioxidant activity of ascorbic acid.

**Figure 15 molecules-27-08728-f015:**
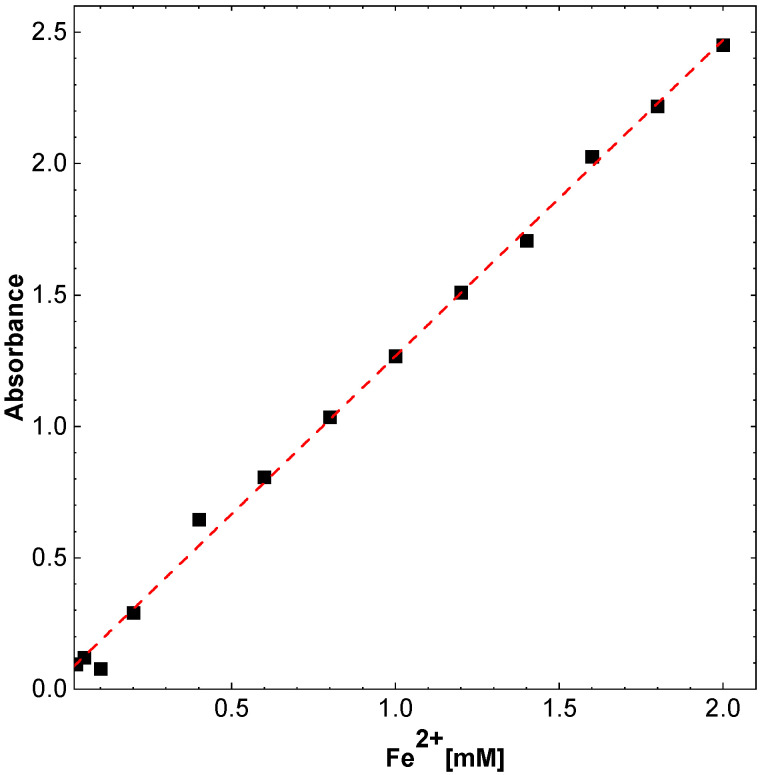
Standard absorbance curve of Fe^2+^.

**Table 1 molecules-27-08728-t001:** Product yield for milk and CDs.

Description (0% Fat)	Difference in Weight (g/L)	Product Yield%
Milk	90.9	-
1 h	34.3	37.7
2 h	34.3	37.7
3 h	35.7	39.3
5 h	35.4	38.9
8 h	27.7	30.5

## Data Availability

Data supporting the reported results are available from the corresponding authors upon written request.

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
