# Peer review of "Milk-Derived Carbon Quantum Dots: Study of Biological and Chemical Properties Provides Evidence of Toxicity"

_molecules, 2022, doi:10.3390/molecules27248728_

Round 1

Reviewer 1 Report

In this manuscript, the authors have prepared carbon dots (CD) with many functional groups by hydrothermal method, and photo luminance spectra are used to show the emission feature. The toxicity of CDs is investigated by the synthesis times from 1 hour to 8 hours, illustrating the toxicity of CDs is highly related to the concentration and time-depended cytotoxic effect. Overall, the manuscript is well prepared and the conclusion can be well supported by the prepared experimental results, therefore, I recommend it can be accepted after minor modification.
1. The peaks at 3300 cm-1 of CDs in Fig 2 of FTIR spectra should not be indexed as the functional group of NH2, but _OH, as the peak is largely right-shifted. The font of C-N is too small to observe and needs to revise.
2. The concentration of Fe2+ in K1h--K8h is not in a linear relationship, is that related to the measurement error or other reasons for it? please discuss it.
3. The HR-STEM images in Figure 4A-B are a lot bit blurry, and is any other way to measure it, such as atomic force microscopy?

Reviewer 2 Report

In the research article entitled “Milk derived carbon quantum dots: Evidence of toxicity through biological, chemical and optical properties studies”, the biological activity milk derived CDs is studied such as Antioxidant and cell viability. The authors have also studied PL and the yield of CDs. but with multiple technical flaws found regarding the important characterizations of CDs such as the Raman spectroscopy and XRD to determine the state of CDs Because the form of carbon materials plays an important role in defining its toxicity. For these reasons, I conclude that the paper should undergo major revision

1.       In the Ultraviolet-visible and Photo–Luminescence Spectroscopy study of CDs from milk, control is missing, i.e, only milk PL and Uv_vis spectra need to be provided like FTIR

2.       In Figure 1, the authors mentioned as K1H, K2h, etc. These abbreviations at the first appearances need to be defined in results as well as figure legend.

3.       Authors need to perform the Raman spectroscopy and XRD to determine the state of CDs like only totally carbon (graphene), graphene oxide, or reduced graphene oxide. Because the state of carbon materials plays an important role in defining their toxicity.

4.       Authors need to update the legend of Figure 5 which is a and B.

5.       In the Antioxidant effect of ascorbic acid and CDs, the concentration of ascorbic acid is in nM and CDs in mg/L. Authors need to represent the units in the same scale (DPPH assay)

6.       Same for FRAP assay as DPPH assay for units of concentration (Query 5)

7.       protocol for Cytotoxicity assay needs to be more detailed in cell density used, and which media used.

8.       In the case of the Cytotoxicity assay, the authors mentioned Staurosporine as a positive control, but in the results section, figure 11 controls seem to be 1% DMSO. Clarify.

9.       Units representation needs to be correct to the standard form like hours to h, etc.

10.   Typographical errors can be avoided. The language and grammar used throughout the manuscript need to be improved. 

Round 2

Reviewer 2 Report

The authors have addressed all the queries and may be accepted for publication